# Advanced System for Enhancing Location Identification through Human Pose and Object Detection

Medrano A. Kevin *,†, Jonathan Crespo †, Javier Gomez † and César Alfaro

Department of Computer Science and Statistics, School of Computer Science and Engineering,
Rey Juan Carlos University, 28933 Madrid, Spain; jonathan.crespo@urjc.es (J.C.); javier.gomez@urjc.es (J.G.);
cesar.alfaro@urjc.es (C.A.)
* Correspondence: 100437357@alumnos.uc3m.es; Tel.: +34-621-390-754
† These authors contributed equally to this work.

**Abstract:** Location identification is a fundamental aspect of advanced mobile robot navigation systems, as it enables establishing meaningful connections between objects, spaces, and actions. Understanding human actions and accurately recognizing their corresponding poses play pivotal roles in this context. In this paper, we present an observation-based approach that seamlessly integrates object detection algorithms, human pose detection, and machine learning techniques to effectively learn and recognize human actions in household settings. Our method entails training machine learning models to identify the common actions, utilizing a dataset derived from the interaction between human pose and object detection. To validate our approach, we assess its effectiveness using a diverse dataset encompassing typical household actions. The results demonstrate a significant improvement over existing techniques, with our method achieving an accuracy of over 95% in classifying eight different actions within household environments.. Furthermore, we ascertain the robustness of our approach through rigorous testing in real-world environments, demonstrating its ability to perform well despite the various challenges of data collection in such settings. The implications of our method for robotic applications are significant, as a comprehensive understanding of human actions is essential for tasks such as semantic navigation. Moreover, our findings unveil promising opportunities for future research, as our approach can be extended to learn and recognize a wide range of other human actions. This perspective, which highlights the potential leverage of these techniques, provides an encouraging path for future investigations in this field.

**Keywords:** computer vision; semantic navigation; machine learning; human pose; object detector; algorithms

## 1. Introduction

The fields of tracking, robot navigation, object detection, and human pose estimation encompass a wide range of applications and research efforts that collectively drive the progress of intelligent systems and robotics. These areas are remarkably diverse in their scope. For example, robot navigation involves the development of algorithms and methods that enable robots to autonomously navigate complex and ever-changing environments, which is proving vital in industries such as logistics and transportation. At the same time, human pose estimation revolves around inferring the skeletal structure and body positions of individuals, contributing to a variety of applications such as gesture recognition, sports analysis, and enhanced human–computer interaction.

Within this area, there are some related works such as [1], which presents a comprehensive review of the use of Unmanned Aerial Vehicles (UAVs), focusing on the development of advanced learning control strategies for improved quadrotor maneuverability. In [2], the authors propose the use of repetitive learning, specifically Iterative Learning Control (ILC) based on optimal approaches, namely Gradient-based ILC and Norm Optimal ILC, to

study the challenges faced by UAVs in monitoring and detecting faults in overhead power lines due to wind disturbances and noise. Another related work is [3], which proposes a distributed controller to address leader–follower consensus for multiple flexible manipulators in the presence of uncertain parameters, unknown disturbances, and actuator dead zones using various control techniques such as adaptive, iterative learning, and sliding mode control.

The focus of our study is primarily on domestic environments, as they present unique challenges and complexities when it comes to understanding human actions based on object interactions. In these environments, the objects play a crucial role in providing valuable information about users' activities and needs. Although the methodology proposed in this work has great potential in the domestic environment, its applicability could be extended to other diverse environments, including factories and shopping centers. In these environments, the method can still effectively interpret human actions, infer user behaviors, and optimize navigation and automation systems.

In domestic environments, the objects in a given area can provide valuable information about the activities taking place and the needs of the users. For example, a person lying on the sofa may indicate that the person is either watching TV or taking a nap. However, understanding how objects can help to identify areas or activities of users in domestic environments can be challenging due to the large number of objects used and the different ways in which they can be combined. Computer vision and machine learning have proven to be useful tools for understanding how the objects can be used to improve navigation systems [4,5], but there are still limitations in their ability to understand the complexity of human movement and interaction with objects.

Pose estimation is also a critical component of our proposed methodology, which must handle real-world variations like lighting and weather. Identifying precise joint coordinates in human pose estimation is especially tough in complex domestic environments. This work presents an innovative perspective on understanding human actions and opens up new possibilities for the design of more intelligent navigation systems. In addition, it is expected that this work will encourage future research in the development of the detection of more than eight human actions trained in this work as well as more accurate and efficient systems for understanding human actions in home environments. The results of this work could have important implications for improving automation and intelligence in smart homes.

The main objective of this work is to develop a solution capable of enhancing the location identification within households for navigation systems. This solution aims to proficiently classify human actions within pre-trained homes using the Python programming language. Data will be collected using a 2D video camera, which will be processed to extract the relevant image features. Different machine learning methods will be developed and evaluated for classification purposes. One crucial aspect of our approach is the careful collection of data at two distinct intervals: 1 s and a total of 6 s for each action. By observing the human pose and the objects present in the interaction environment during these specific time intervals, our method effectively captures essential temporal information for action learning.

As a result, our proposed methodology presents a robust framework for learning and classifying human actions based on the observation of poses and the objects present in the interaction environment during specific time intervals. Tests have been conducted attempting to classify areas as a room based solely on the presence of objects. However, a room can be used for many common activities, such as studying or eating, so it was decided to conduct tests by separating the data to see if it was possible to classify zones in a home based solely on the presence of objects. In the results section, the importance of not only using the human pose but also considering the detected objects in the environment is observed to achieve accurate classification. The combination of these components increases the accuracy and efficiency of the action classification process, making our solution a valuable contribution to the field of human action recognition in home environments.

The rest of this paper is organized as follows. Section 2 provides an overview of related work on object, activity, and human pose recognition methods. In Section 3, we describe our proposed methodology to support indoor location identification using machine learning methods combining human pose and object detection. Section 4 presents the results of experiments and performance evaluation. Finally, we summarize some discussions and main conclusions from this research in Section 5.

## 2. Related Works

The different robotic applications used in this new era of data digitization are increasingly advanced thanks to the integration of machine learning and deep learning algorithms, which are sub-elements of what is known today in a general sense as artificial intelligence [6]. These areas of research are constantly evolving with applications in various fields, including understanding the usefulness of objects in home environments [7,8].

### 2.1. Classification Techniques for Activities in Homes

One of the most common approaches in research is the use of deep learning techniques for object and activity classification in home environments. For example, the work of [9] uses convolutional neural networks for object and activity identification in home environments, achieving an accuracy of 92%. The article mentions some common challenges in applying deep learning techniques for activity recognition, such as the need for large amounts of labeled data and the need to adapt to different contexts and environments. Specific limitations of certain deep learning techniques for activity recognition are also described, such as the lack of ability of convolutional neural network models to model temporal dynamics. Another common approach is the use of rule-based learning techniques, as in the work of [10] that presents a prototype for an optical fall detection system based on pose estimation, running in real time, which uses various approaches, such as machine learning, deep learning, and a rule-based algorithm for detecting falls. All these methods achieve an accuracy exceeding 94% when tested on publicly available datasets. However, machine learning shows limitations in real-time applications due to the scarcity of available training data. On the contrary, the rule-based approach demonstrates a greater capacity for generalizing different types of fall events.

A non-invasive activity recognition system for home environments using small, easy-to-install, and low-cost sensors is presented in [11]. The results of the experiments conducted in this study indicate that the system can detect relevant activities such as toileting, bathing, and grooming with varying accuracy rates between 25% and 89%. A novel approach to self-supervised sensor representation learning for smartphone-based human activity recognition is presented in [12]. The method uses a multi-task temporal convolutional network to identify potential transformations in raw input data. The results obtained outperform supervised methods and surpass traditional unsupervised techniques such as autoencoders. The study suggests that with further refinements, this approach could further bridge the gap between unsupervised and supervised feature learning.

In addition to the aforementioned approaches, there are other works that employ similar methods for identifying objects and their use in specific activities. For instance, the work of [13] uses object tracking and trajectory analysis techniques for identifying object usage patterns in home environments. Another interesting work is that of [14], which supplies a concrete understanding of the variant sensing principles of image and video processing, segmentation, feature extraction, classification, cross-validation, and sensitivity analysis techniques. These techniques were combined to implement and evaluate a human activity recognition system based on multiple modalities.

### 2.2. Object Detectors and Object Trackers

In recent years, different object detection algorithms have been developed that have significantly improved the accuracy and speed of real-time detection. One of the most popular algorithms is YOLO (You Only Look Once), which has improved in different

versions over the years, with version 3 achieving high performance in terms of accuracy and speed [5], while YOLO v4 focuses on further improving accuracy and speed [15]. Another popular algorithm is SSD (Single Shot Multibox Detector), which is a real-time object detector that uses a single iteration of a neural network to predict the presence of multiple objects in an image. SSD has proven to be one of the most efficient techniques for real-time object detection [16].

The researchers in [17] propose an application for robot navigation in indoor environments that integrates a YOLO v3 convolutional neural network (CNN) for furniture and household object detection into the simultaneous localization and mapping (SLAM) algorithm. The proposed application allows automatic SLAM map generation and the simultaneous detection of rooms in unknown environments. Similarly, the authors in [18] present a real-time computer vision-based object detection and recognition framework to improve indoor robot navigation. The proposed system uses SLAM for navigation and YOLO for object detection, resulting in lower computational requirements and lower network weight.

A vision system for object detection and recognition in indoor environments is proposed in [19]. This system uses Support Vector Machines, RGB, and depth images, combined with several segmentation techniques and feature extraction methods based on geometric shape descriptors and a bag of words. The author in [20] presents a novel approach to improve the efficiency of object localization, which incorporates a dynamic Bayesian network designed to automatically detect and update the state of an object based on human activity, thereby improving the efficiency of object localization, which is particularly beneficial for service robots in domestic environments.

Another important factor to achieve the objectives of this work is the use of object-tracking algorithms. One of the most effective approaches for this task is object trackers based on Kalman filtering, which use a dynamic model of the object to estimate its position and velocity over time [21]. In recent years, deep learning has been used to improve the accuracy and speed of object tracking, using CNN to detect the object in each frame and then using a tracking model to track the object over time [22].

*2.3. Human Pose Detectors*

Human pose detection algorithms aim to detect and estimate human poses in an image or video. This problem has been considered in computer vision for decades, but recent advances in machine and deep learning techniques have led to significant improvement. Human pose detectors based on deep learning are usually divided into two main categories: regression methods and detection methods. Regression methods estimate a person's pose through a neural network, while detection methods use object detection techniques to locate and estimate the pose of each person in an image.

One of the most prominent approaches is described in [23], which proposes a deep neural network for human pose estimation from 2D images. Another interesting proposal consists on a human pose detection approach that uses Part Affinity Fields to estimate the pose of multiple people in real time, improving accuracy and performance speed [24]. The authors in [25] propose two methods to improve pose-sensitive detection and human pose retrieval systems based on convolutional neural network features, offering superior performance over previous methods.

Google has developed MediaPipe, which is a popular and powerful open-source tool for real-time human pose detection. The library offers a variety of pre-trained models and tools for the custom development of pose detection models, and among the most prominent pre-trained models is the full-body pose detection model "BlazePose" [26], which uses a deep convolutional neural network and a particle-based pose estimation architecture to detect human body poses in real time.

Also, deep learning algorithms are used for detecting human pose. The authors in [27] compare Convolution Neural Network–Long Short-Term Memory (CNN-LSTM) networks with other models such as Multilayer Perceptron (MLP), Long-term Recurrent

Convolutional Networks (LRCN) or LSTM for classifying the dance pose by excerpted salient details attaining high-performance results, up to 98%, in some metrics as accuracy, precision, recall, AUC or F1 score. The author in [28] uses a CNN model called the quaternion field pose network (qfiled PoseNet) to detect the pose of objects from a single aerial image with good results, as demonstrated in experiment on the DOTA1.5 and HSRC2016 datasets. At the same time, the authors in [29] present PoseFormer, a purely transformer-based approach for 3D human pose estimation in videos without convolutional architectures involved, which achieves state-of-the-art performance on two benchmark datasets, Human3.6M and MPI-INF-3DHP, according to extensive experiments.

In general, these studies have highlighted the promising potential of computer vision, machine learning, and deep learning for understanding human actions within domestic environments. Building upon this groundwork, our research presents a groundbreaking machine learning-based methodology to better comprehend human actions through the analysis of human movement and object interactions in home environments. Our approach aims to greatly improve the accuracy and efficiency of understanding human actions, opening doors to more effective and reliable applications in this field.

## 3. Methodology

The proposed methodology utilizes a combination of machine learning and computer vision techniques for understanding human actions in home environments. The approach is based on human movement and object analysis to improve the accuracy and efficiency of semantic classification of home areas.

### 3.1. Description of the General Framework

In this section, we present an overview of the core features of the modules in Figure 1 to facilitate understanding the proposed methodological framework.

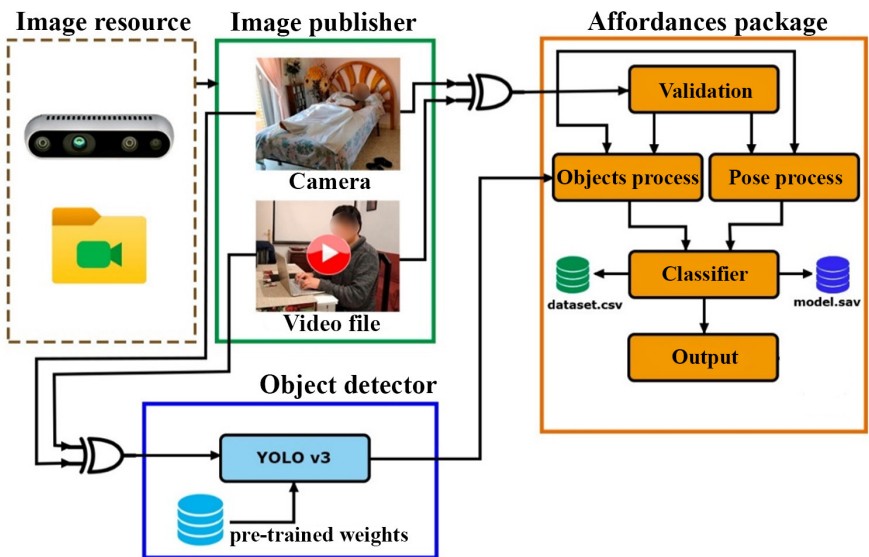

**Figure 1.** General framework.

The affordances module forms the central component of our methodology and facilitates the processing of object and human pose data as well as the execution of classification tasks. It consists of five sub-modules:

- Validation: This sub-module ensures the correct reception of video data and validates the presence of at least five human pose data points to indicate the feasibility of data collection.
- Objects process: The *objects process* sub-module allows for the collection and management of data related to different objects detected in the environment.

- Pose process: The *pose process* sub-module is responsible for the collection and management of human pose data.
- Classifier: The *classifier* sub-module supports functionalities for the training, evaluation, and validation of classification methods.
- Output: The *output* sub-module is where the final classification result is reproduced or obtained.

The image resource and image publisher modules facilitate the process of retrieving video data. Two types of resources can be used to retrieve video data: a camera or a video file in MP4 format. These modules are responsible for providing the video data to the affordances and the object detector modules, either from the camera or from a video file. The connection between modules is shown with a symbol that represents an XOR logic gate, indicating that only one of the input elements is valid and not both.

The object detector module is responsible for managing the detection of objects in the video resource provided by the image publisher module and subsequently sending the detection results to the affordances module. The object detector identifies objects present in the environment, which is essential for understanding human–object interactions during action recognition.

### 3.2. Procedures of the Methodology

The proposed methodology consists of five sequential steps, as illustrated in Figure 2, which are required to enhance location identification through human pose and object detection.

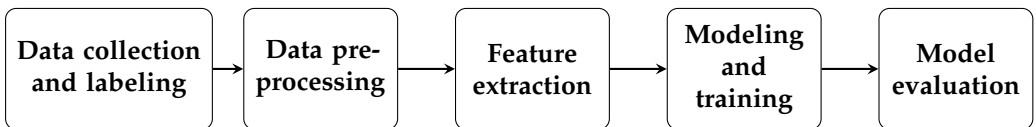

**Figure 2.** Steps of the proposed methodology: (1) Data collection and labeling, (2) Data preprocessing, (3) Feature extraction, (4) Modeling and training, (5) Model evaluation.

Data collection

The developed system implements a supervised data collection process, storing the data in a standardized format using text files. Data collection occurs according to the configuration shown in Figure 1, either through video recordings or camera input. For instance, to gather data for the "lying on sofa" action, multiple instances of the same action with different human poses are recorded, keeping the camera focused on the action area without any movement. Data capture occurs at one-second intervals, starting from the first second and concluding at the sixth second. During this period, the data are supervisedly labeled to assign the corresponding action. To ensure balanced representation, a sufficient amount of data has been collected for each action, with approximately the same number of examples for all actions.

Data preprocessing

The dataset is processed to remove incomplete or incorrect data, and then, it is segmented into specific human actions for further analysis.

As observed in Figure 3, the full visibility of all pose points is not always achievable. However, the library includes calculations within its model to determine values of $(x, y, z)$ even when the visibility is less than 0.5.

Figure 4 illustrates the representation of data for different human pose points. These representations show examples of a person doing different activities: lying (Figure 4a) or sitting (Figure 4b) on a sofa, and performing exercises (Figure 4c). The acquisition of 3D points is achieved using the MediaPipe library, where the input image is a 2D RGB image. Various angles and distances have been considered for data collection, with the camera height fixed at 1.3 m.

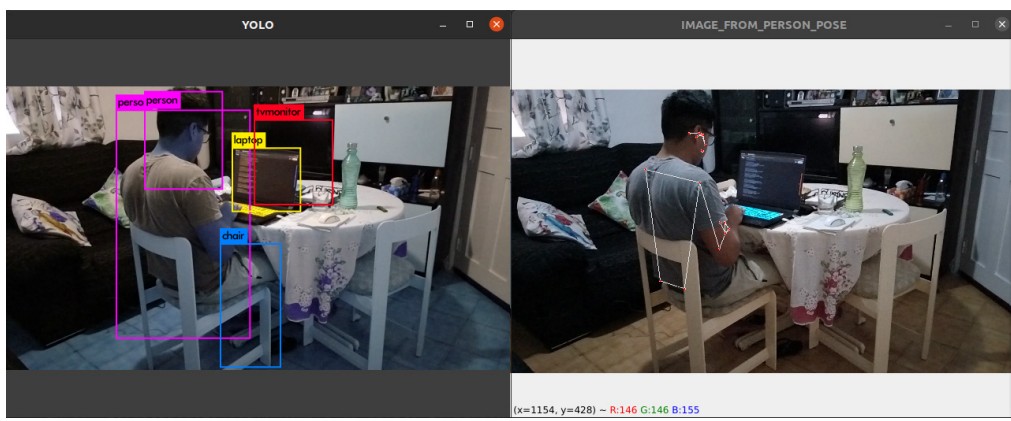

**Figure 3.** Data collection and labeling.

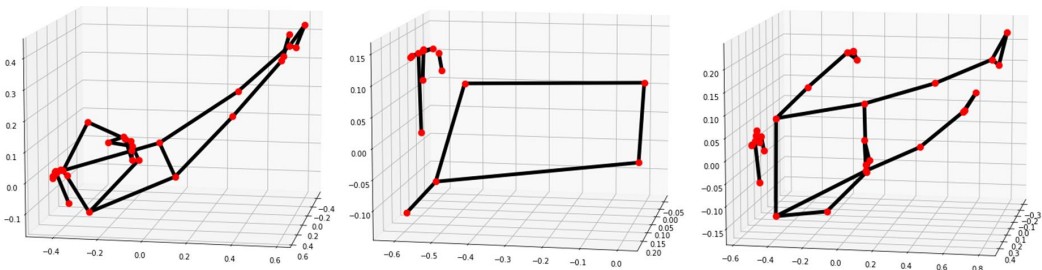

(**a**) Representation of a human pose lying on a sofa.

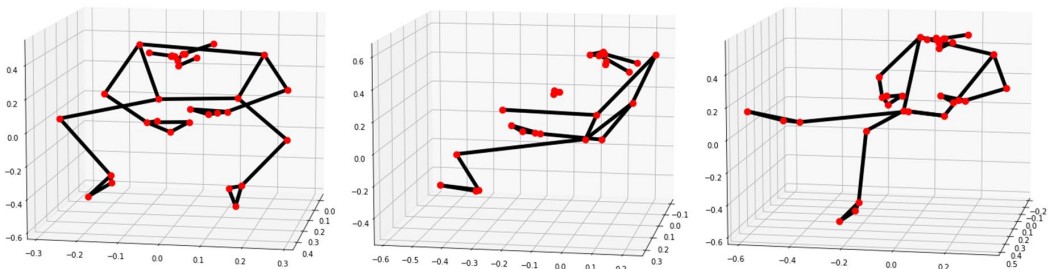

(**b**) Representation of a human pose sitting on a sofa.

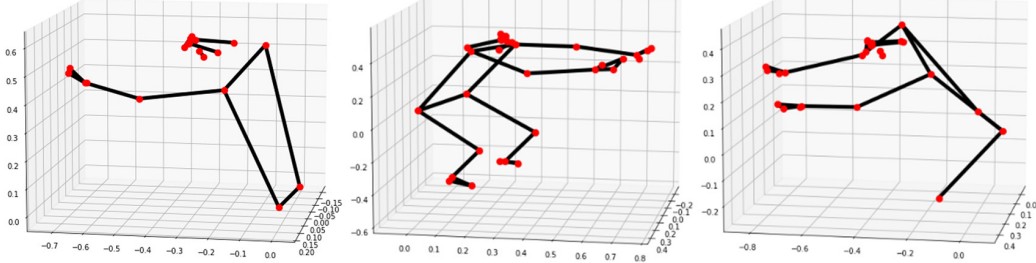

(**c**) Representation of a human pose working out.

**Figure 4.** Different representations of human pose.

Feature extraction

Feature extraction techniques will be used to obtain relevant information about objects and human pose.

Figure 5 depicts the structure of a category sample, consisting of 6 data subsections representing the attributes of the category at different time intervals, each separated by one second, encompassing a total of 6 s. (1) These data are from human poses obtained through the MediaPipe library. The content has been structured using a subset of only 9 specific

points, which was a decision made after careful consideration to reduce computational time. These 9 points were selected after conducting extensive data reduction experiments. The criteria for selecting these points were based on identifying those that best represent the most relevant information of the human pose for the classification task. It is important to note that a total of 33 points can be obtained, but the use of 9 points was considered sufficient to achieve optimal performance in the classification process. These obtained data are already scaled and are represented in 3D position data from a 2D image, spatial position values $(x, y)$, and visibility value z. (2) The presented data correspond to the objects detected using the YOLO v3 object detector, which utilizes the official COCO object names list, comprising 80 categories. However, for this particular version, an exhaustive selection of only 58 categories has been performed, representing objects commonly found in an average household. Examples of these objects include sofas, beds, chairs, tables, and televisions, among others, which are commonly encountered in a typical home. This rigorous selection process has allowed us to focus the analysis on the most relevant elements for object detection applications in home environments. (3) This value corresponds to the category that describes the human action being represented. For instance, one of the trained actions is "lying on bed". In total, eight different output classes are being worked on: reading a book, using a laptop, lying on sofa, sitting on sofa, lying on bed, drinking with a cup, working out and playing a console.

It is essential to clarify that human pose estimation can be achieved either in 2D or 3D, with the primary difference lying in the desired type of output result. With the 2D output, we receive a visual that resembles a stick figure or skeleton representation of the various key points on the body. While with 3D human pose estimation, we receive a visual representation of the key points on a 3D spatial plane, with the option of a three-dimensional figure instead of its 2D projection. For this study, the 2D model is established first, and then the 3D version is lifted from that visual.

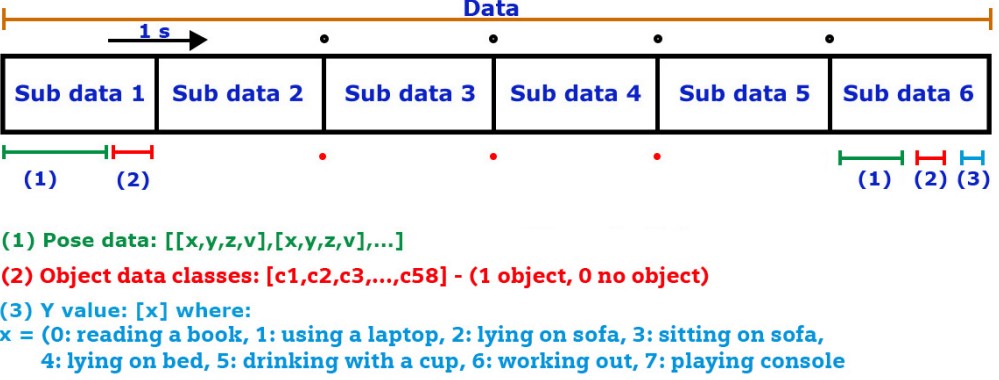

**Figure 5.** Data structure.

Modeling and training of machine learning algorithms

Machine learning methods will be developed to comprehend human actions based on the characteristics obtained from human pose and objects detected. A wide range of machine learning algorithms have to determine which would be the most effective for the classification task.

- Support Vector Machine (SVM): SVMs [30] can be used for binary classification, multiclass classification, and regression tasks. The choice of kernel type, kernel coefficient, and regularization parameter can have a significant impact on the performance of the model. For example, a linear kernel may be appropriate for linearly separable data, while a non-linear kernel like the radial basis function (RBF) may be better suited for non-linearly separable data. The regularization parameter C controls the tradeoff between maximizing the margin and minimizing the classification error and can be tuned to optimize performance.

- Gradient Boosting (GB): GB [31] is a powerful ensemble method that can be used for classification and regression tasks. The learning rate determines the contribution of each individual tree to the final model, and a smaller learning rate can help prevent overfitting. The maximum depth of the trees and the number of estimators (trees) can also be tuned to optimize performance.
- Extreme Gradient Boosting (XGB): XGB [32] is a popular variant of gradient boosting that is known for its speed and performance. In addition to the hyperparameters mentioned above for gradient boosting, XGBoost also includes regularization parameters like L1 and L2 regularization as well as a subsampling ratio parameter that controls the fraction of observations used to train each individual tree.
- Light Gradient Boosting Machine (LGBM): LGBM [33] is another variant of gradient boosting that is designed for efficient performance on large datasets. In addition to the hyperparameters mentioned above for gradient boosting, LightGBM also includes hyperparameters like the number of leaves per tree and the minimum gain to split a node that can be used to optimize performance.
- K-Nearest Neighbors (K-NN): K-NN [34] is a simple but effective non-parametric method that can be used for classification and regression tasks. The number of neighbors and the distance metric used to compute distances between points are the two main hyperparameters that can be tuned to optimize performance. A larger number of neighbors can help prevent overfitting, while different distance metrics like Euclidean distance or cosine distance may be more appropriate depending on the dataset and problem being solved.

Table 1 summarizes the main hyperparameters of each method and the values that have been used for each of them in the experiments conducted in this study.

**Table 1.** Classification methods and hyperparameters used.

| Method | Hyperparameters |
| --- | --- |
| Support Vector Machine (SVM) | Kernel type: RBF<br>Kernel coefficient: 0.001<br>Regularization C: 1 |
| Gradient Boosting (GB) | Learning rate: 0.1<br>Maximum depth: 3<br>Criterion: friedman mse<br>Number of estimators: 100 |
| XGBoost (XGB) | Objective: multi:softprob<br>Booster: gbtree<br>Maximum depth: None<br>Number of estimators: 100 |
| LightGBM (LGBM) | Learning rate: 0.1<br>Maximum depth: −1<br>Number of leaves: 31<br>Number of estimators: 100<br>Boosting type: gbdt |
| K-Nearest Neighbors (K-NN) | Number of neighbors: 5<br>Metric: minkowski<br>Leaf size: 30<br>Weights: Uniform |

Model evaluation

The accuracy and efficiency of the machine learning method will be evaluated using the test data. Additionally, the hyperparameter optimization process was applied after obtaining the initial results. To evaluate the performance of the proposed framework, the concept of the confusion matrix [35] is used. Let $n$ be the number of different classes; a

confusion matrix of size $n \times n$ associated with a classifier shows the actual and predicted classification values.

Table 2 illustrates a $2 \times 2$ confusion matrix in which each cell has a specific interpretation as follows:

- $t_p$: indicates the number of positive instances classified accurately.
- $f_p$: is the number of actual negative instances classified as positive.
- $f_n$: indicates the number of actual positive instances classified as negative.
- $t_n$: is the number of negative instances classified accurately.

**Table 2.** Confusion matrix.

|  | **Predicted Positive** | **Predicted Negative** |
|---|---|---|
| Actual Positive | $t_p$ | $f_n$ |
| Actual Negative | $f_p$ | $t_n$ |

The confusion matrix provides the basis for obtaining various performance measures [36,37]. In this study, the following metrics are used to evaluate the performance of the machine algorithms.

The accuracy metric is a measure commonly used to evaluate the performance of a classification model. It represents the proportion of correctly classified instances out of the total number of instances in a dataset. The accuracy of a classification model is calculated using the following equation:

$$Accuracy = \frac{t_n + t_p}{t_n + f_p + f_n + t_p}$$

The recall metric, also known as sensitivity or true positive rate, measures the ability of the model to correctly classify instances of a given class out of all the instances that truly belong to that class. The recall of a classification model is calculated using the following equation:

$$Recall = \frac{t_p}{t_n + f_n}$$

The precision metric is a performance measure that assesses the accuracy of a model's predictions for each class. It measures the proportion of correctly classified instances for a given class out of the total number of instances predicted to be in that class. The recall of a classification model is calculated using the following equation:

$$Precision = \frac{t_p}{t_p + f_p}$$

The $F_1$ metric is a performance measure that provides a balanced assessment of the model's performance by taking into account both the precision and the recall for each class. It is the harmonic average of the precision and recall, where an $F_1$ score reaches its best value at 1 (perfect precision and recall) and worst at 0. Therefore, this score takes both false positives and false negatives into account. The $F_1$ score of a classification model is calculated using the following equation:

$$F_1 = \frac{2t_p}{2t_p + f_p + f_n}$$

The Matthews Correlation Coefficient (MCC) is a performance measure that quantifies the quality of predictions in multi-class classification tasks. It takes into account the $t_p$, $t_n$, $f_p$, and $f_n$ for each class and calculates a correlation coefficient that ranges between $-1$ and $+1$, where values of $+1$, $0$, and $-1$ indicate an accurate prediction, a random prediction and

a mismatch between the predicted and actual classes, respectively. The MMC measure of a classification model is calculated using the following equation:

$$\mathrm{MMC} = \frac{t_p \times t_n - f_p \times f_n}{\sqrt{(t_p + f_p)(t_p + f_n)(t_n + f_p)(t_n + f_n)}}$$

The Area Under the Curve (AUC) is a metric derived from the Receiver Operating Characteristic (ROC) curve, which plots the $t_p$ rate against the $f_p$ rate for different classification thresholds. In multi-class classification, the AUC metric is typically calculated by averaging the pairwise comparisons between each class and the rest. It represents the probability that a randomly selected instance from one class will be ranked higher by the model than a randomly selected instance from another class by the model. The AUC ranges from 0 to 1, where a higher value indicates better discrimination and overall classification performance.

Cohen's kappa coefficient ($\kappa$) is a statistical measure used to assess the degree of agreement between the predictions of a multi-class classification model and the true class labels. It measures the agreement between the predicted class labels and the true class labels, taking into account both correct predictions and misclassifications. A kappa coefficient of 1 represents a perfect agreement, while a coefficient close to 0 indicates no better than chance agreement.

### 3.3. Experimental and Testing Areas

The experiments were carried out in three different types of homes, focused on areas such as bedrooms, living rooms, study areas, dining rooms, exercise areas, rest rooms, and game rooms (Table 3).

**Table 3.** Areas classes.

| Class | Area | Human Action |
|-------|------|--------------|
| 0 | Reading room | Reading a book |
| 1 | Study room | Using a laptop |
| 2 | Rest room | Lying on sofa |
| 3 | Living room | Sitting on sofa |
| 4 | Bedroom | Lying on bed |
| 5 | Dining room | Drinking with a cup |
| 6 | Exercise area | Working out |
| 7 | Game room | Playing console |

Table 3 provides an overview of different classes along with their corresponding areas and associated human actions. Figure 6 shows some of the images from the recordings of the different actions. As mentioned before, they were recorded from different angles and distances without changing the height of 1.3 m.

The proposed methodology is designed to handle challenges in human pose estimation, including robustness to real-world variations such as lighting and weather. Special attention is given to capturing complex body positions by utilizing carefully collected data and feature extraction techniques. The system is designed with control mechanisms to handle suddenly interrupted actions, ensuring data validity. Additionally, the proposed methods can be adapted to handle multiple persons and their interactions.

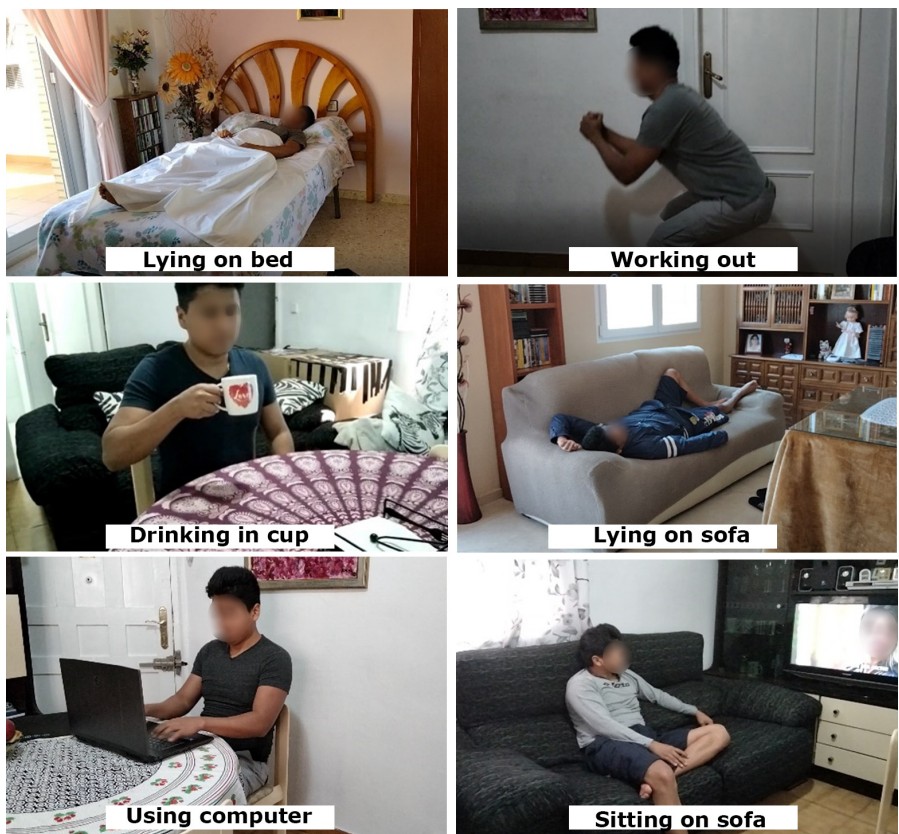

**Figure 6.** Examples of recordings of a person performing different actions.

## 4. Results and Experiments

In this section, we describe the experiments and results conducted to evaluate our methodology. For this research, it was decided to conduct three experiments in order to detect house areas: (i) using only objects of interest; (ii) using only the human pose data; and (iii) combining the use of objects of interest and the human pose data. The aim is to assess the importance and relevance of these types of data when applying a machine-learning method for classification. By examining these data separately, we seek to understand the individual impact of each on the classification process. This approach allows for a clearer and more precise understanding of how each type of data contributes to the model's performance and its ability to classify effectively.

It is important to note that our proposed system is designed to capture a person's range of motion within the context of single-person interaction scenarios. The training data used for the machine learning models includes a wide variety of body positions and movements to ensure that the system can accurately detect different human actions. However, the system can be adapted to multiple persons as well by incorporating additional data for multi-person scenarios.

For each experiment, we report a table summarizing the average of each of the performance metrics and the computation time, as well as the confusion matrices of the methods with the best and worst classification results.

Table 4 provides a description of the object data, the human pose data, and the combined object and human pose data. It includes information about the target type, the number of columns and rows in each dataset, the number of features, and the number of folds used for cross-validation. Note that for each experiment, the dataset was divided into training and testing subsets. Specifically, 70% of the data was used for training, while the remaining 30% was used as a testing set.

**Table 4.** Object, pose, and a combination of object and pose data description.

| Description | Object | Human Pose | Object and Human Pose |
|---|---|---|---|
| Target | Multi-class | Multi-class | Multi-class |
| Dataset | rows: 1606, columns: 343 | rows: 1603, columns: 217 | rows: 1601, columns: 559 |
| Train set | rows: 1124, columns: 343 | rows: 1122, columns: 217 | rows: 1122, columns: 559 |
| Test set | rows: 482, columns: 343 | rows: 481, columns: 217 | rows: 481, columns: 559 |
| Features | 342 | 216 | 558 |
| Folds | 10 | 10 | 10 |

### 4.1. Classification Using Objects of Interest

In this section, we present the results obtained solely using object data, considering the same data structure. These results are of paramount importance, as they provide a detailed understanding of the influence of objects on the performance of the machine learning method. By analyzing only the object data, we can gather specific information regarding their relevance and contribution to the accurate classification of instances. This investigation offers a comprehensive and insightful perspective on the pivotal role played by object data in our analysis.

Figure 7 shows the confusion matrices obtained from the SVM (Figure 7a) and k-NN (Figure 7b) methods, with the true class labels on the x-axis and the class predictions on the y-axis. The first diagonal contains the correct classifications, while all other entries show misclassifications. As can be seen, the largest errors come from misclassifying classes 0 (reading room) and 7 (game room). For example, in the confusion matrix of Figure 7a, it can be seen that 22 instances of class 0 are incorrectly predicted as class 7, while 7 and 17 instances of class 7 are incorrectly predicted as class 0 and 6 (exercise room), respectively. Similar results are observed in the confusion matrix of Figure 7b.

The results of the experiments are summarized in Table 5, which shows the average of the performance metrics for each of the five classification methods evaluated and the average computation time in seconds. As we can see, GB demonstrates the highest accuracy of 72.97%, which is followed closely by XGB at 72.53% and LGBM at 72.08%. These methods exhibit competitive performance in terms of AUC, with GB achieving the highest value of 95.08%.

**Table 5.** Performance comparison using only objects in the environment.

| Method | Accuracy | AUC | Recall | Prec. | $F_1$ | $\kappa$ | MCC | Time (s) |
|---|---|---|---|---|---|---|---|---|
| GB | 0.7297 | 0.9508 | 0.7297 | 0.7512 | 0.7286 | 0.6910 | 0.6941 | 1.0510 |
| EGB | 0.7253 | 0.9488 | 0.7253 | 0.7394 | 0.7251 | 0.6859 | 0.6881 | 0.1990 |
| LGBM | 0.7208 | 0.9469 | 0.7208 | 0.7360 | 0.7202 | 0.6809 | 0.6832 | 0.7050 |
| K-NN | 0.6754 | 0.9078 | 0.6754 | 0.6935 | 0.6690 | 0.6290 | 0.6337 | 0.1580 |
| SVM | 0.7297 | 0.9508 | 0.7297 | 0.7512 | 0.7286 | 0.6910 | 0.6941 | 0.1470 |

Regarding the recall metric, GB again outperforms the other methods with a score of 72.97%. This indicates its ability to correctly identify positive cases. However, it is important to note that all the methods have similar recall scores, suggesting reasonable performance across the board.

Precision analysis reveals that GB achieves the highest score (75.12%), indicating its superior ability to avoid false positives compared to the other methods. Similarly, GB achieves the highest $F_1$ score (72.86%), which balances precision and recall, demonstrating its robust performance. In addition, when evaluating the $\kappa$ and MCC metrics, GBC consistently outperforms the other methods, indicating its overall superiority.

Finally, although GB has a remarkable performance, it is important to consider the trade-off between performance and computation time. GB has a longer computation time of 1.0510 s compared to the other methods, while K-NN has the shortest computation time of 0.1580 s.

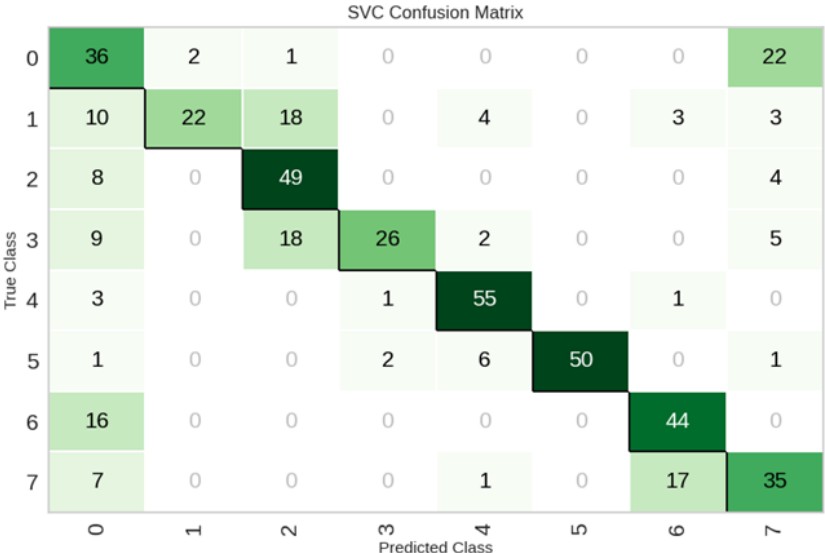

(**a**) Confusion matrix from SVM method using objects.

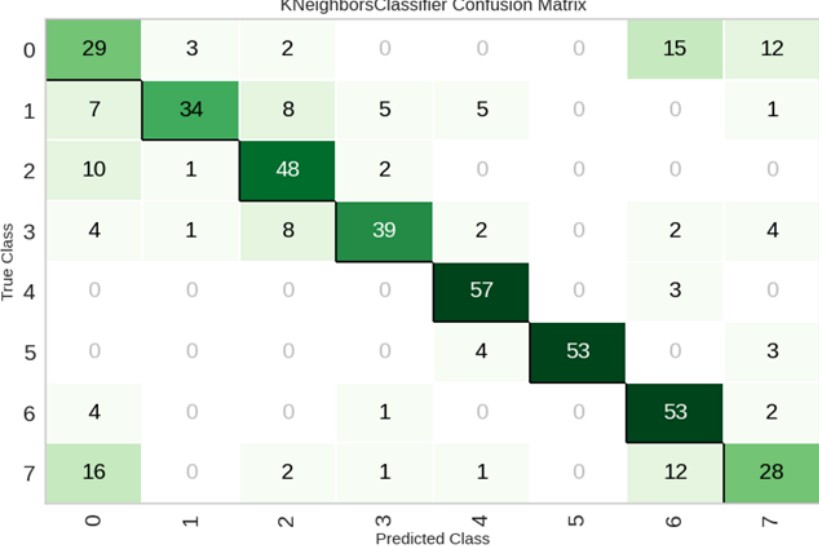

(**b**) Confusion matrix from K-NN method using objects.

**Figure 7.** Examples of confusion matrices using objects of interest.

### 4.2. Classification Using Human Pose Data

In this section, we present the results obtained solely using human pose detection data. These findings are significant because they provide a comprehensive understanding of the influence of human pose on the performance of the machine learning method. By focusing solely on the human pose recognition data, we gain specific insights into its relevance and contribution to achieving accurate instance classification.

Figure 8 shows the confusion matrices obtained from the LGBM (Figure 8a) and SVM (Figure 8b) methods. On the one hand, as can be seen from the confusion matrix in Figure 8a, the LGBM method shows a low rate of misclassifications. On the other hand, the analysis of the confusion matrix obtained with the SVM method (Figure 8b) shows that the classes 0 (reading room), 4 (bedroom), and 7 (game room) are the most frequently misclassified.

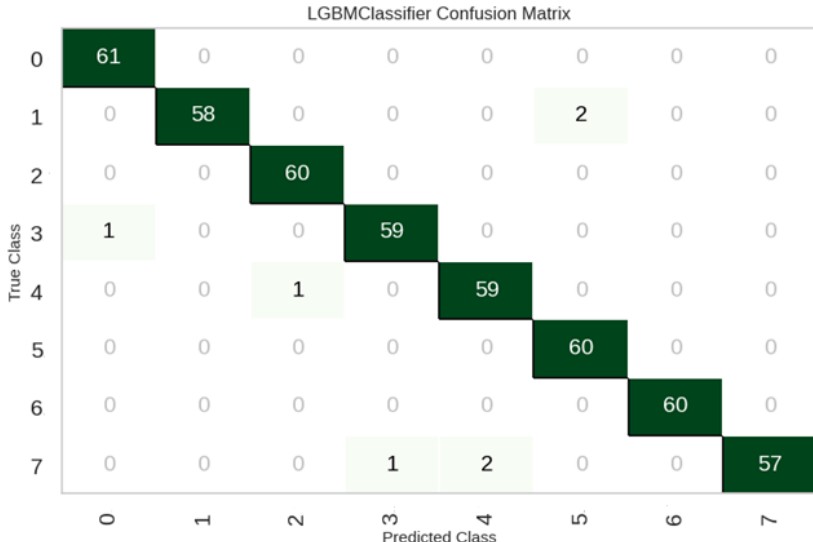

(**a**) Confusion matrix from LGBM method using human pose.

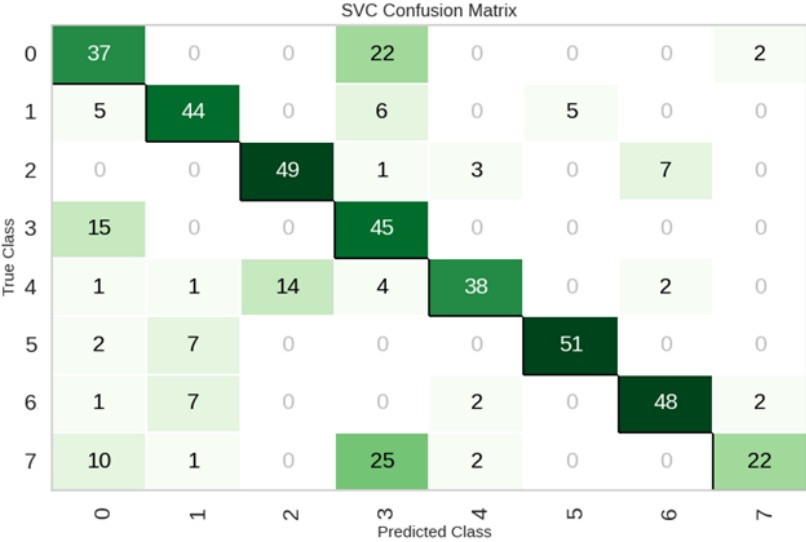

(**b**) Confusion matrix from SVM method using human pose.

**Figure 8.** Examples of confusion matrices using human pose.

The average of the performance metrics and the average computation time in seconds are summarized in Table 6. As we can see, LGBM has the highest accuracy with 97.86%, which is closely followed closely by XGB at 96.52% and GB at 95.99%. These methods exhibit exceptional performance in accurately classifying instances.

**Table 6.** Performance comparison using human pose.

| Method | Accuracy | AUC | Recall | Prec. | $F_1$ | $\kappa$ | MCC | Time (s) |
|---|---|---|---|---|---|---|---|---|
| LGBM | 0.9786 | 0.9992 | 0.9786 | 0.9799 | 0.9785 | 0.9755 | 0.9757 | 12.8920 |
| EGB | 0.9652 | 0.9984 | 0.9652 | 0.9673 | 0.9648 | 0.9602 | 0.9606 | 9.1760 |
| GBM | 0.9599 | 0.9986 | 0.9599 | 0.9634 | 0.9597 | 0.9541 | 0.9547 | 51.5040 |
| K-NN | 0.9251 | 0.9909 | 0.9251 | 0.9308 | 0.9239 | 0.9144 | 0.9155 | 0.2540 |
| SVM | 0.6809 | 0.9445 | 0.6809 | 0.7274 | 0.6734 | 0.6353 | 0.6445 | 0.2770 |

In terms of the AUC, which measures the ability of the model to discriminate between classes, the LGBM method achieves the highest value at 99.92%, which is closely followed by XGB at 99.84% and GB at 99.86%. These high AUC values indicate the methods' excellent discrimination capability.

The recall metric, which measures the ability of the models to correctly identify positive instances, shows consistently high scores for all models, with the LGBM method achieving the highest recall at 97.86%, which is closely followed by XGB at 96.52% and GB at 95.99%.

Precision analysis reveals that the LGBM method achieves the highest precision score (97.99%), indicating its superior ability to avoid false positives compared to the other methods. Similarly, LGBM achieves the highest $F_1$ score (97.85%), which balances precision and recall, demonstrating its robust performance. In addition, the LGBM classifier consistently outperforms the other methods when assessing $\kappa$ and MMC metrics, indicating its overall superiority in terms of agreement and correlation.

Finally, although the LGBM outperforms the other methods in terms of performance metrics, it is worth noting that the computation time for LGBM is significantly higher compared to the other methods with a value of 12.8920 s, while K-NN has the shortest computation time of 0.2540 s.

### 4.3. Classification Combining Human Pose and Objects of Interest

In this section, we present the results obtained by considering both object data and human pose, using the same data structure as previously presented. The analysis of these results reveals the fundamental importance of incorporating both object and human pose data to enhance classification models. This integration of information provides a more comprehensive and accurate understanding of the instances to be classified, enabling superior performance in machine learning methods. By combining object and human pose information, relevant features and patterns crucial for classification are captured and utilized, leading to a substantial improvement in the accuracy and effectiveness of the models. These findings highlight the crucial relevance of including object and human pose data in the analysis and enhancement of classification algorithms.

Figure 9 shows the confusion matrices obtained from the LGBM and SVM classification methods. As can be seen from the confusion matrix in Figure 9a, the LGBM method performs well. On the contrary, the SVM method (Figure 9b) obtains several classification errors, with the highest number of errors when classifying the instances of class 7, where 38 out of 60 instances are misclassified.

Table 7 shows the classification results and the average computation time in seconds when combining the environmental objects and the human pose. As we can see, among the methods, LGBM achieved the highest accuracy (98.66%), AUC (99.99%), and $F_1$ score (98.65%). It also showed high recall, precision, $\kappa$, and MCC values, indicating its effectiveness in correctly classifying instances.

**Table 7.** Performance comparison combining human pose and environmental objects.

| Method | Accuracy | AUC | Recall | Prec. | $F_1$ | $\kappa$ | MCC | Time (s) |
|--------|----------|--------|--------|--------|--------|--------|--------|----------|
| LGBM | 0.9866 | 0.9999 | 0.9866 | 0.9876 | 0.9865 | 0.9847 | 0.9849 | 13.9950 |
| GB | 0.9768 | 0.9996 | 0.9768 | 0.9783 | 0.9767 | 0.9735 | 0.9737 | 56.8530 |
| EGB | 0.9723 | 0.9996 | 0.9723 | 0.9741 | 0.9722 | 0.9684 | 0.9687 | 14.4640 |
| K-NN | 0.9250 | 0.9896 | 0.9250 | 0.9343 | 0.9259 | 0.9143 | 0.9156 | 0.3240 |
| SVM | 0.8277 | 0.9824 | 0.8277 | 0.8482 | 0.8275 | 0.8031 | 0.8062 | 0.8160 |

Similarly, the GB and XGB methods also performed well with accuracy scores of 97.68% and 97.23%, respectively. These methods showed comparable results in terms of AUC, recall, precision, $F_1$ score, $\kappa$, and MCC.

In contrast, the K-NN and SVM methods achieved relatively lower accuracy scores of 92.50% and 87.05%, respectively. These methods demonstrated lower performance in terms of AUC, recall, precision, and $F_1$ score compared to the boosting methods.

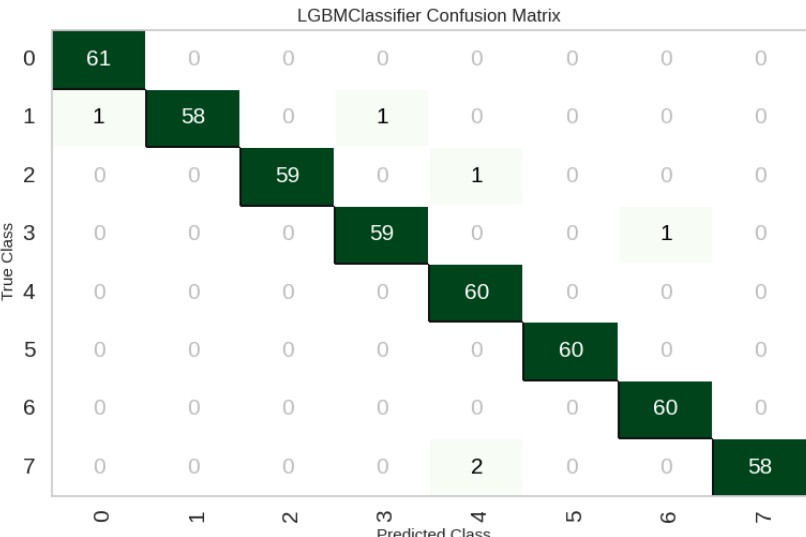

(**a**) Confusion matrix from LGBM method combining objects and human pose.

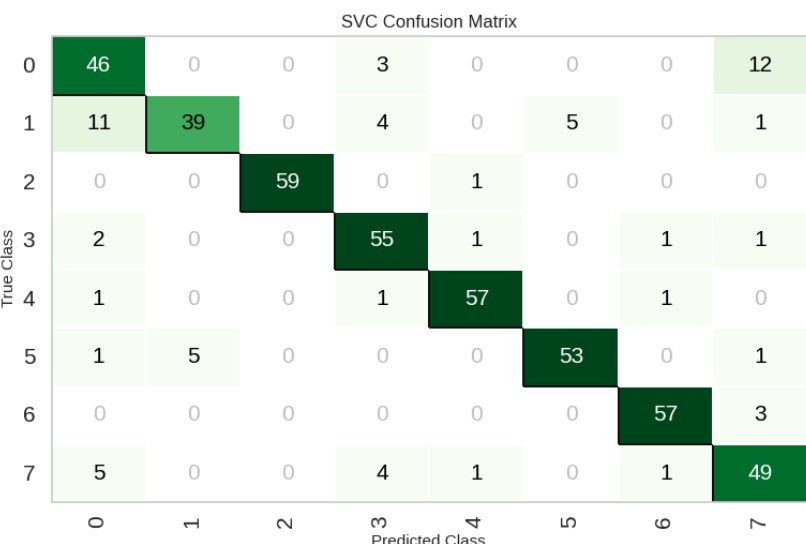

(**b**) Confusion matrix from SVM method combining objects and human pose.

**Figure 9.** Examples of confusion matrices combining objects and human pose.

Finally, Table 8 shows the accuracy comparison between the three different analyses. It is worth noting that including both object data and human pose data in the analysis led to improved classification results for all methods except the K-NN algorithm. The combination of object and human pose features allowed for a more comprehensive understanding of the data and improved the models' ability to accurately classify instances. In this table, it can be observed that when both sources of information, the detected objects, and the detected human pose, are used, the precision of the methods ranges from a maximum of 0.9866 to a minimum of 0.8277, with an average of 0.9368. However, if only the information from the detected objects is considered, the precision of the methods reaches a maximum of 0.7297 and a minimum of 0.6754. In this case, the average precision is 0.71618, which is significantly lower than the average precision obtained with the combination of both sources of information. The same trend applies to the precision obtained from the human pose data alone. With a maximum precision of 0.9786 and a minimum of 0.6809, the average precision is 0.90194, which is also lower than the average precision when using both sources of information.

**Table 8.** Accuracy Comparison of Machine Learning Methods.

| Method | Objects | Human Pose | Objects and Human Pose |
|---|---|---|---|
| LGBM (LightGBM) | 0.7208 | 0.9786 | 0.9866 |
| GB (Gradient Boosting) | 0.7297 | 0.9599 | 0.9768 |
| EGB (Extended Gradient Boosting) | 0.7253 | 0.9652 | 0.9723 |
| K-NN (K-Nearest Neighbors) | 0.6754 | 0.9251 | 0.9250 |
| SVM (Support Vector Machine) | 0.7297 | 0.6809 | 0.8277 |

## 5. Conclusions and Discussion

This study demonstrates that the inclusion of both object data and human pose data has proven beneficial in improving the classification results of various models. The combination of these two sources of information enables achieving a precision of up to 0.9866 in the best-performing model, surpassing the classification achieved when only one of them is utilized. However, it is worth noting that employing a greater number of classes than the ones used in this study (eight) might not yield equally successful results.

From the results obtained, it can be concluded that the combination of object data and human pose data has certain advantages. Object data can be used to classify rooms, but there are cases where it does not provide enough information to distinguish between two different rooms that may contain the same type of object. The average accuracy in experiments using only object data was 0.71618. The movement of people, as seen through human pose, can also be used to classify different rooms, as people perform different actions in different places. However, there are also cases where the actions are similar and the classification may fail. The average accuracy of the experiments using only human pose information was 0.90194. Combining both types of information allows us to classify the rooms more accurately. The average accuracy in this case was 0.9368.

While the majority of the models benefited from the inclusion of both object and human pose data, it is noteworthy that the K-NN algorithm did not show the same level of improvement. This could be attributed to the specific characteristics of the K-NN algorithm, which relies heavily on distance-based metrics for classification. It is possible that the inclusion of human pose data, which represents spatial relationships rather than direct object attributes, did not align well with the distance-based nature of the K-NN algorithm. Further investigation and experimentation with different algorithms specifically tailored to handle both types of data may be warranted to fully explore their potential synergies.

The results and methodology presented in this study could serve as a starting point for future research in the field of semantic navigation planning for robots by understanding human actions for the identification of location in home environments.

In addition, it is important to highlight that the system's versatility extends beyond single-person interactions. The methodology can be adapted and extended to handle multi-person scenarios, offering potential applications in various settings such as factories and shopping centers. Future research efforts can explore techniques to incorporate multi-person pose estimation, further enhancing the system's capabilities and enabling it to cater to diverse user scenarios. Furthermore, the proposed methodology addresses challenges related to robustness in dealing with sudden interruptions by carefully collecting data at specific time intervals to effectively capture essential temporal information for action learning.

**Author Contributions:** Conceptualization, M.A.K. and J.C.; Data curation, M.A.K.; Formal analysis, M.A.K., J.C., J.G. and C.A.; Investigation, M.A.K.; Methodology, M.A.K., J.C., J.G. and C.A.; Software, M.A.K.; Supervision, J.C.; Validation, J.C., J.G. and C.A.; Visualization, M.A.K.; Writing—original draft, M.A.K.; Writing—review and editing, J.C., J.G. and C.A. All authors have read and agreed to the published version of the manuscript.

**Funding:** This research received no external funding.

**Institutional Review Board Statement:** Not applicable.

**Informed Consent Statement:** Not applicable.

**Data Availability Statement:** Not applicable.

**Conflicts of Interest:** The authors declare no conflict of interest.

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
