# Peer review of "Advanced System for Enhancing Location Identification through Human Pose and Object Detection"

_machines, doi:10.3390/machines11080843_

Round 1
Reviewer 1 Report
This paper presents an analysis of machine learning-based object detection in the context of high-level mobile robot navigation systems and analyzing the use of data from visual characteristics of human posture and objects in real-world environments to perform the semantic navigation. In my opinion, the methodology is good and the presentation of both the equations and some of the results of this paper is relatively clear. My detailed comments are as follows: -
Point 1: For the abstract, it should be rewritten more precisely to include the main ideas and actual contributions of the manuscript. The abstract contains only general ideas about the mobile robot navigation systems and training machine learning which does not fully cover what was accomplished in the manuscript. This should be presented in an interesting way.
The part of this work has been done in a conference paper due to that, try a) to highlighted not only your proposed methos but also your extended work, add this in the introduction section to let reader follow your work, b) I prefer to highlight the contributions in introduction section as a bullet point.
Point 2: Please check the coherence of this line's statements. " In general, these works have demonstrated the potential." page 4.
Check typos on page 5 and where you found in rest of manuscript.
Point 3: Is it important to add literary comparisons in the introduction section explaining why the authors are pushed towards common household activities. Knowing that there are areas that are more important than others, such as factory workers activities, and athlete pose detection in sports from professional athletics to amateur competitions. Please indicate the different areas of application in the introduction.
Point 4: It is important to suggest adding new and recent works to the list of sources to show the special diversity within the scope of the tracking, robot navigation, object detection and pose. Accordingly, it is possible to add systems that carry a new direction, such as iterative Learning methods within the introduction paragraph, and indicate the extent of the difference between them and the proposed system from these recently published examples in the high-level mobile robot navigation systems such as,
a- “An Advanced Unmanned Aerial Vehicle (UAV) Approach via Learning-Based Control for Overhead Power Line Monitoring: A Comprehensive Review"
b- “Application of Norm Optimal Iterative Learning Control to Quadrotor Unmanned Aerial Vehicle for Monitoring Overhead Power System"
c- "Distributed control of multiple flexible manipulators with unknown disturbances and dead-zone input".
Point 5: For the references, it is preferable to replace some of them because they are old research and the results presented in some of these articles are unrealistic.
Point 6: Check typos in the conclusion.
Point 7: It is important to answer this question clearly.
Generally, pose estimation can be achieved either in 2D or 3D, with the primary difference lying in the desired type of output result. With the 2D output, we receive a visual that resembles a stick figure or skeleton representation of the various key points on the body. While with 3D human pose estimation, we receive a visual representation of the key points on a 3D spatial plane, with the option of a three-dimensional figure instead of its 2D projection. From manuscript, the 2D model is established first, and then the 3D version is lifted from that visual.
For this point of view, is your proposed system have ability to perform the following: -
1) Pose estimation needs to be robust to challenging real-world variations such as lighting and weather. Therefore, it is challenging for image processing models to identify fine-grained joint coordinates.
2) Does your proposed system within the algorithms can capture a person's in complete range of motion, so more complex body positions will be detected correctly. Or you only train some data via machine learning based.
3) Did the proposed methods have a challenge in case dealing with suddenly interrupted.
4) Do the proposed methods with (machine learning-based) have ability to deal and train data with a multi person pose within interaction environment.

I did not check the plagiarism completely (please check because some work, but not all, was done in the conference documents)
Reviewer 2 Report
This research work propose training machine learning methods to recognize the visual characteristics of human posture and objects in their interaction environment in order to recognize associated utilities. The paper is very interesting, well structrured and well written.
I suggest to the researchers to change the title becuase it is a little bit long.
Reviewer 3 Report
This paper is written in a very interesting contest, however authors must improve a lot the manuscript to be published. Some suggestions are:
1. The title, abstract and introduction refer to robot navigation, however this subject was not considered in the model nor in results.
2. In related works, authors must cite works related with CNN + LSTM or Transformers neural networks which are widely used to detect human pose in a sequence of images.
3. In Figure 1, it is not clear how the object detection and the pose detection is used in robot navigation.
4. In line 178, authors said "the semantic robot navigation", But, the results are not being used in robot navigation.
5. In section 3.2, Data Collection sub-section, Authors must clarify: how many labeled images or poses were collected? was the dataset balanced? what the authors did to balance the dataset?
6. In line 202 and 203, why 9? why not 10? what was the criteria to consider those 9 points?
7. In line 207, authors said "58 classes", which ones?
8. In figure 5, it is not clear what are those sub-data parts.
9. In lines 246 and 247, Authros should be careful in this subject, since, depending on the application and dataset, optimization of the hyperparameters should be considered. Then, this optimization will be done before or after this comparison?
9. At the end of section 3.2, It apparently detects the pose or the objects, but the model proposed is not coherent with the title neither with the abstract, it is not used in robot navigation neither combine both sources of data as described in the introduction.
10. In line 296, authors said "the object of the cup is used", only one object? what about dishes? glasses?
11. In lines 396 and 397, It is not clear how authors used the same structure to fuse two types of input data. In addition, the model proposed does not consider from the beginning this situation.
I think it is an interesting work, but it must be improved.
My best,
